# Metatranscriptomic investigation of single *Ixodes pacificus* ticks reveals diverse microbes, viruses, and novel mRNA-like endogenous viral elements

Calla Martyn,[1,2] Beth M. Hayes,[1,3] Domokos Lauko,[1] Edward Midthun,[4] Gloria Castaneda,[5] Angela Bosco-Lauth,[4] Daniel J. Salkeld,[6] Amy Kistler,[5] Katherine S. Pollard,[2,5,7] Seemay Chou[1,5]

**ABSTRACT**   Ticks are increasingly important vectors of human and agricultural diseases. While many studies have focused on tick-borne bacteria, far less is known about tick-associated viruses and their roles in public health or tick physiology. To address this, we investigated patterns of bacterial and viral communities across two field populations of western black-legged ticks (*Ixodes pacificus*). Through metatranscriptomic analysis of 100 individual ticks, we quantified taxon prevalence, abundance, and co-occurrence with other members of the tick microbiome. In addition to commonly found tick-associated microbes, we assembled 11 novel RNA virus genomes from *Rhabdoviridae*, *Chuviridae*, *Picornaviridae*, *Phenuiviridae*, *Reoviridae*, *Solemoviridae*, *Narnaviridae* and two highly divergent RNA virus genomes lacking sequence similarity to any known viral families. We experimentally verified the presence of these in *I. pacificus* ticks across several life stages. We also unexpectedly identified numerous virus-like transcripts that are likely encoded by tick genomic DNA, and which are distinct from known endogenous viral element-mediated immunity pathways in invertebrates. Taken together, our work reveals that *I. pacificus* ticks carry a greater diversity of viruses than previously appreciated, in some cases resulting in evolutionarily acquired virus-like transcripts. Our findings highlight how pervasive and intimate tick–virus interactions are, with major implications for both the fundamental biology and vectorial capacity of *I. pacificus* ticks.

**IMPORTANCE** Ticks are increasingly important vectors of disease, particularly in the United States where expanding tick ranges and intrusion into previously wild areas has resulted in increasing human exposure to ticks. Emerging human pathogens have been identified in ticks at an increasing rate, and yet little is known about the full community of microbes circulating in various tick species, a crucial first step to understanding how they interact with each and their tick host, as well as their ability to cause disease in humans. We investigated the bacterial and viral communities of the Western blacklegged tick in California and found 11 previously uncharacterized viruses circulating in this population.

**KEYWORDS**   arthropod vectors, vector-borne disease, metagenomics, endosymbionts, nonhuman microbiome, innate immunity, bioinformatics, metatranscriptomics

Ticks are increasingly important disease vectors for humans and livestock, particularly in the United States, where they account for more cases of vector-borne diseases than mosquitoes. Approximately fifty thousand confirmed cases of tick-borne diseases are reported annually (1), which is likely an underestimate due to diagnostic challenges associated with Lyme disease and our poor understanding of rare tick-borne diseases or diseases of unknown etiology. Currently, the majority of field surveillance studies

Address correspondence to Katherine S. Pollard, kpollard@gladstone.ucsf.edu, Seemay Chou, seemay.chou@ucsf.edu, or Amy Kistler, amy.kistler@czbiohub.org.

Seemay Chou is an employee of Arcadia Sciences.

See the funding table on p. 16.

of tick-associated microbes focus on *Borrelia burgdorferi,* the causative agent of Lyme disease, and a select number of other known human pathogens, such as *Rickettsiae* (causative agents of Rocky Mountain spotted fever and other Rickettsioses), *Anaplasma phagocytophilium* (the etiologic agent of Anaplasmosis), and Powassan virus. Although the full diversity of microbes carried by ticks is much greater than those definitively linked to human disease (2–22), we know strikingly little about the ecology or disease implications of most tick-associated microbes.

Furthermore, we are only beginning to appreciate the broader role microbes play in tick biology. Like many other invertebrates, tick–microbe interactions go far beyond the transmission of human pathogens. Tick–microbe interactions can be antagonistic, neutral, or beneficial (23–26). Some microbes, such as the bacterial genus *Rickettsia*, play fundamental roles in tick physiology through stable and symbiotic interactions (26). It is not as well known whether ticks also form stable, symbiotic interactions with viruses and how such interactions may influence tick biology. Some viruses have been identified not only in live ticks but also in many laboratory-passaged tick cell lines (27). Microbes have also been implicated in shaping the evolution of ticks through the horizontal transfer of bacterial genes (28) and endogenization of viral sequences as an immune response (29–32).

The relationship between ticks and viruses is a subject of much interest. Ticks are able to carry viruses from a broad range of families, some of which have been identified as causative agents of disease in humans such as tick-borne encephalitis virus and Powassan virus of the *Flaviviridae* family of positive sense RNA viruses, and severe fever with thrombocytopenia syndrome virus (33) of the *Phenuiviridae* family of multi-segmented negative sense RNA viruses. While field microbiome studies have increased our catalog of tick-associated viruses (33), a number of experimental strategies and technical hurdles have limited the scope and depth of microbiome analyses in ticks. First, field studies often sequence pooled tick samples, preventing quantitative examination of microbial prevalence, co-occurrence, and per-sample relative abundance. These metrics could enable more sophisticated analyses of transmission dynamics and ecology. Furthermore, our ability to capture lower-abundance microbes is hampered by the dominance of tick host sequences in metatranscriptomic libraries. This limits our understanding to the most abundant microbes in ticks, which does not necessarily coincide with all microbes that have important roles in disease or tick physiology.

Moreover, our understanding of tick microbiota in North America is currently biased towards tick host species historically associated with human diseases, such as *Ixodes scapularis*, the primary vector for Lyme disease in the Eastern United States. However, in recent years, there has been an expansion of tick-borne disease cases on the West Coast of the United States that have been attributed to other, less-studied tick vector species. *Ixodes pacificus* is a major tick species extending from Northern Mexico to British Columbia (34). *I. pacificus* ticks are most abundant in California, where they cover 96% of all counties (35) and are responsible for the majority of human tick bites (36). They are vectors for a variety of well-characterized human pathogens such as *B. burgdorferi*, *Borrelia miyamotoi*, *Babesia odocoilei*, *Bartonella* spp., *A. phagocytophilum*, and *Ehrlichia* spp. (37). Despite this, *I. pacificus* is substantially understudied compared to the eastern black-legged tick *I. scapularis*.

To provide much-needed insight into tick-borne microbes in the western United States, we examined the microbiomes of *I. pacificus* ticks collected from two coastal habitats in California where humans are likely to encounter them (36, 38, 39). In order to capture lower-abundance microbes, we coupled an experimental microbial enrichment workflow with RNA sequencing to profile both bacteria and RNA viruses. Analysis of microbiomes at the level of individual ticks enabled us to also quantify patterns of microbial prevalence and abundance. We performed follow-up laboratory-controlled experiments examining microbial localization within tick compartments and across tick life stages, which provided additional insights into potential transmission dynamics and the symbiotic nature of tick-virus relationships.

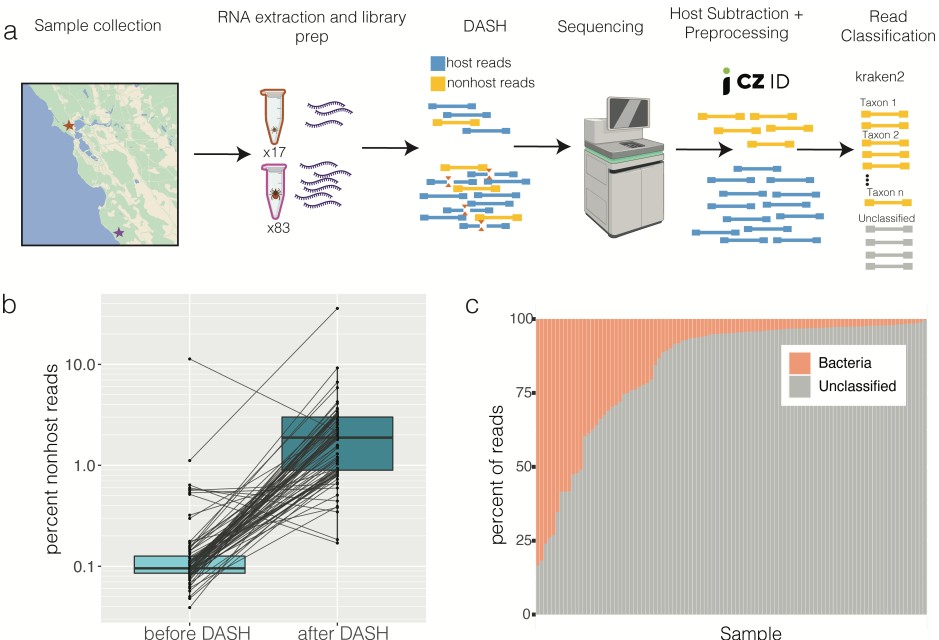

**FIG 1** *Ixodes pacificus* metatranscriptomics experimental approach. (a) Eighty-three adults and 17 nymphs were collected from Garrapata State Park (purple) and China Camp State Park (red), respectively. RNA was extracted from whole bodies of individual *I. pacificus* ticks, sequencing libraries were prepared, and depletion of abundant sequences by hybridization (DASH) was performed to deplete abundant tick sequences. After sequencing, reads were quality filtered and tick-derived reads were removed using CZID. The remaining reads were classified by kraken2. Map was drawn based on an image from Google Maps. (b) The percent of non-host reads as classified by CZID for matched libraries before and after DASH. Lines connect individual libraries. (c) Percentage of non-host reads classified per sample by kraken2.

## RESULTS

### Establishment of an RNA-based approach to defining composition of field tick microbiomes

We set out to define the metatranscriptome of *I. pacificus* ticks collected from coastal California, focusing on two sites associated with human exposure (36–40). We examined the two most developmentally advanced life stages (nymphal and adult) that are more amenable to single-tick sequencing due to greater individual biomass. These sample sets included adult ticks from Garrapata State Park and nymphal ticks from China Camp State Park. Adults were collected in the winter (2018) and nymphs were collected in the spring (2019) so that we could investigate seasons and life stages enriched for human contact. In total, RNA libraries were sequenced for 100 individual ticks.

The majority of whole-tick RNA libraries are composed of tick ribosomal RNA, which reduces the power to detect lower-abundance sequences with homology to bacteria and viruses. To address this challenge, we enriched microbial sequences by experimentally depleting abundant tick sequences through depletion of abundant sequences by hybridization (DASH) (Fig. 1a) (41). For adult tick libraries, DASH-based depletion enriched non-host sequences nearly 10-fold (Fig. 1b). Sequencing libraries generated for smaller nymphal ticks were not of sufficient concentration to effectively perform DASH. After sequence quality filtering and removal of host sequences, the remaining non-host sequences were classified using the metagenomic classifier kraken2 (42) (Fig. 1a). The samples varied substantially in the proportion of non-host reads that could be classified, ranging from 0.2% to 83% (Fig. 1c). All classified reads by kraken2 were bacterial; no known viruses were identified. Because of this, unclassified sequences were analyzed with a custom pipeline that enabled the detection of divergent viruses (see below, Materials and Methods).

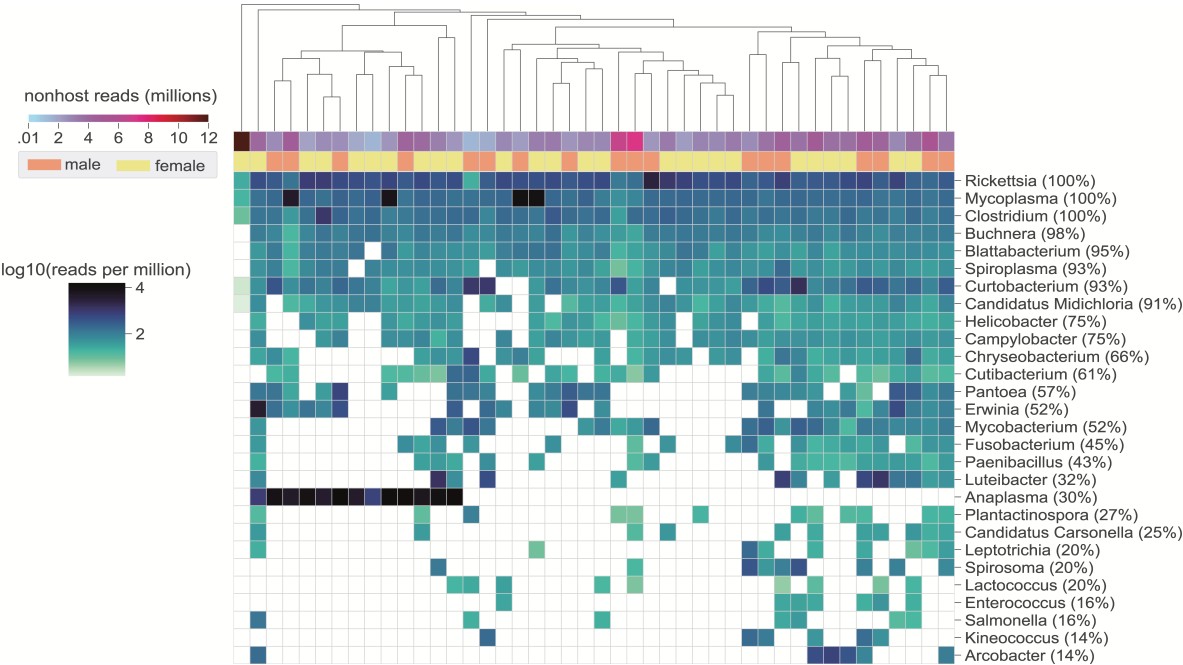

**FIG 2** Bacterial genera detected in *Ixodes pacificus*. Heatmap showing reads per million (rpm) of bacterial genera as classified by kraken2. Plot is limited to samples with at least 1 million non-host reads and genera detected in at least five samples. Prevalence in the selected samples (rows) is shown next to the genus name, and genera are ordered by decreasing prevalence. Samples (columns) are hierarchically clustered using Euclidean distance.

To assess the general validity of our approach to characterizing the microbiota of field ticks, we first quantified the bacterial component of the tick metatranscriptomes. We identified 114 bacterial genera across the data set with a median of 11 genera per tick (Fig. S1a). Larger libraries generally had more classified genera, though there was substantial variability in the relationship between sequencing depth and detected richness (Fig. S1a and S2). We compared the taxonomic composition of our samples with previously reported tick-associated human pathogens, such as *Rickettsia, Anaplasma*, and *Coxiella*, which are most commonly linked to *I. pacificus* ticks (3, 6, 15, 37). *Borrelia*, *Borreliella*, *Ehrlichia*, and *Bartonella* are also human pathogens known to circulate in this species, although typically at lower frequencies. *Ehrlichia*, *Borrelia,* and *Borreliella* were identified at rates of 2–3% across the full data set but not in samples with a more stringent quality cutoff of at least 1 million non-host reads (Fig. 2; Fig. S2). We did not identify *Coxiella*, *Bartonella*, or *Francisella* in any ticks, indicating they are either absent in this population or present at levels too low to be detected.

Of the samples with at least 1 million non-host reads, several genera we identified were found at frequencies similar to previous reports (3, 6, 15, 37). We detected the endosymbiont *Rickettsia* in 100% and *Anaplasma* in approximately 30% of ticks (Fig. 2). We also observed several instances of coinfections between *Ehrlichia*, *Borrelia*, *Borreliella*, and *Anaplasma* species (Fig. S1b). In total, results from our bacterial analyses are largely consistent with previously reported analyses of bacterial constituents of tick microbiota, suggesting our RNA-based approach to characterizing tick-associated microbes can indeed be applied to field studies. These results gave us confidence that our sequencing and analysis pipeline could reliably report on microbiomes of field *I. pacificus* ticks at nymphal and adult life stages.

## Optimized workflow enables detection of low-abundance bacteria and RNA viral transcripts

Our combined approach of host depletion and RNA-sequencing opened up several unique lines of inquiry. In addition to known tick-borne human pathogens, we identified

several bacterial genera previously undetected in *I. pacificus*. Although these genera had high prevalence across our samples, they were present at low relative abundances and likely escaped detection in previous studies in the absence of DASH-based microbial enrichment. *Mycoplasma*, a genus that has been linked to Lyme-like disease in patients with tick exposure (43, 44), was identified in all of the selected libraries (Fig. 2). *Blattabacterium*, *Buchnera*, *Spiroplasma*, and *Candidatus Midichloria* were all present in at least 90% of samples, and *Candidatus Carsonella* was identified in 25% of samples (Fig. 2). To our knowledge, none of these known endosymbionts have been commonly identified in *I. pacificus*, and this is the first report of *Blattabacterium*, *Buchnera*, and *Candidatus Carsonella* in any tick (45–47).

Sequencing individual ticks also provided sufficient resolution for co-occurrence analyses. We assessed whether the presence of one microbial genus increases the statistical likelihood that another microbial genus will be present in the same tick host (48). This revealed 14 pairs of bacterial genera detected together in a statistically significant number of samples (Fig. S3). Of note, there was a strong positive association between the endosymbiont *Candidatus Carsonella* and *Chryseobacterium*, a genus that has been shown to be pathogenic to soft ticks but tolerated by hard ticks (49). The remaining statistically significant co-occurrence relationships did not include any microbial genera known to be tick-associated, and we hypothesized that they may be environmental contaminants, such as soil bacteria (14), that were detectable despite our rinsing of the ticks and use of water controls. Therefore, outside of the *Candidatus–Chryseobacterium* case, we did not find clear evidence that any microbes associated with *I. pacificus* actively promote the colonization or growth of other microbes.

Our RNA-based sequencing approach also enabled the identification of both known and previously unidentified RNA viruses in *I. pacificus* ticks. Ticks are known to carry a diversity of viruses, and the majority of known transmissible arboviruses have RNA genomes (15, 50–54). We sought evidence of these and any novel viruses in our metatranscriptomes. To do so, we first developed a bioinformatics strategy because standard tools for microbiome analysis (e.g., kraken2 [42]) did not detect any known tick viruses in our data. This is a common phenomenon in RNA virus discovery, due to the diversity of RNA viruses not being well represented in reference databases. In keeping with other viral discovery efforts (55–57), we searched for sequences containing an RNA-dependent RNA polymerase (RdRp) domain using HMMER (58) (Fig. 3a).

Using this strategy, we detected a total of 13 distinct putative viral genomes in our *I. pacificus* field specimens and determined their prevalence across the data set as well as their relative abundance within each sample (Fig. 3b, Table S1). For each putative viral genome, we recovered was provisionally named based on the geographical features in the region in which the samples were collected. Underscoring the novelty of these putative viral genome sequences, the vast majority (11/13) showed less than 80% amino acid identity to their nearest relative in the NCBI non-redundant protein database (Fig. 3b). Despite this, phylogenetic analysis of their RdRp amino acid sequences using NCBI/ICTV methodology (Materials and Methods, Fig. S8) mapped most sequences to a viral order (*Bunyavirales* and *Mononegavirales*) or a viral family (*Rhabdoviridae*, *Picornaviridae*, and *Narnaviridae*) that has been previously shown to have members infecting ticks. An exception corresponds to two partial putative viral genome sequences that we recovered, Kasler Point virus and Wildcat Canyon virus. These divergent sequences showed very low amino acid identity to their best RdRp matches and were provisionally assigned to the recently discovered ormycovirus clade, a group of unclassified RNA viruses that has not been phylogenetically placed within any existing order or family of viruses. The remaining two viral genome sequences we recovered, Notley's Landing virus and Rocky Ridge virus, showed >90% amino acid identity to previously described members of the *Solemoviridae* and *Chuviridae* families that were isolated from ticks (Table S1; Fig. S8). A detailed summary of the attributes of each of these viruses is provided in the Supplementary Note.

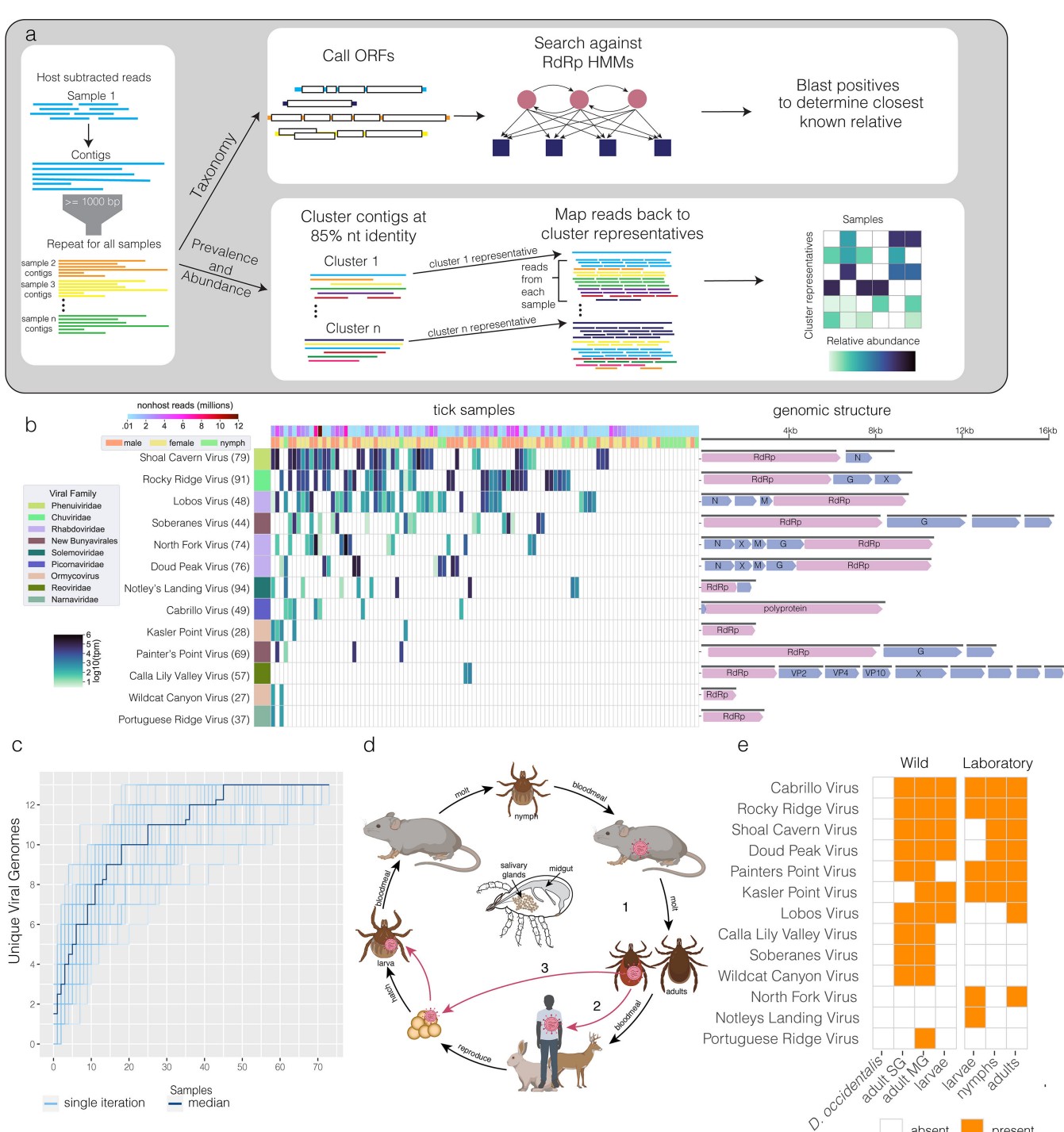

**FIG 3** Discovery of novel viral sequences in *Ixodes pacificus*. (a) Analysis pipelines for identification of viruses: reads were assembled into contigs and open reading frames were predicted. Resulting protein sequences were scanned for the presence of RNA-dependent RNA polymerase (RdRp) using HMMER and classified by viral family using closest BLAST hits and taxonomically annotated following NCBI/ICTV methodology. Prevalence and abundance were determined by clustering contigs and mapping reads to the cluster representatives. (b) Heatmap of transcripts per million (tpm) of each identified viral RdRp-containing contig across the data set. Average amino acid identity of the aligned region to closest known sequence is displayed to the right of the virus name. Open reading frames are displayed to the right of the heatmap and annotated with identified proteins (RdRp, G = glycoprotein, N = nucleoportein, M = matrix protein, VP = viral protein). Open reading frames with "X" contain homology to know viral proteins of unknown function, and those with no annotation had no identified homology to known proteins. (c) Rarefaction plot showing increase in number of viral genomes for each sample analyzed. Samples were randomly shuffled 50 times, and the median was plotted in dark blue. (d) Schematic of *I. pacificus* three-stage life cycle. Viruses can be transmitted horizontally between hosts and ticks (Continued on next page)

**FIG 3** (Continued)

(1 and 2) or vertically from adult females to their offspring (3). (e) Table summarizing detactinon of each viral contig across tick samples. SG = salivary glands, MG = midgut.

## RNA viruses are stable constituents of tick microbiota

Our RNA-based approach to tick microbiome characterization led to the discovery of highly divergent members of several viral families in *I. pacificus* field ticks. To follow up on these results, we next asked if the 13 distinct viral sequences we detected were likely to represent the full virome of the sampled ticks or only the most abundantly transcribed strains. To do so we performed rarefaction analysis, which shows the number of new taxa discovered as a function of the number of samples sequenced. Our rarefaction curve appears to be approaching an asymptote, and the estimated true number of viruses in this population using the Chao index is 14.2 (Fig. 3c). These results indicate that we have likely discovered the majority of viruses in this population with a median of 1.7 million non-host sequences after DASH host-subtraction. This is equivalent to an overall sequencing depth of 106 million reads given our median amount of host depletion. We also observed that relatively few samples are needed to saturate viral discovery for a given population at this sequencing depth.

We also performed co-occurrence analysis with our newly characterized tick viromes. Not only did we identify a broad diversity of viral genomes, but we also found evidence of co-occurrence within individual ticks. Ticks had a median of two viral species detected, with a maximum of 6 in one individual (Fig. S4a). We found a statistically significant positive relationship between Portuguese Ridge virus (*Narnaviridae*), Wildcat Canyon virus (*ormycovirus*), and Kasler Point virus (*ormycovirus*) (Fig. S3). Notably, all three of these viral genome assemblies contained only an RdRp; no additional segments or genes were identified. Since Narnaviruses are single-gene ribonucleoprotein complexes lacking structural proteins or capsids, it is possible that the two viruses of unknown origin replicate and transmit in a similar manner (see Supplemental Note: *Narnaviridae* and Ormycovirus). Our findings support the model that ticks can harbor multiple viral species per individual, suggesting that RNA viruses are consistent and stable members of the tick microbiome.

## Life-stage and tissue tropism of viruses

Arthropods are known to tolerate viral infection more easily than vertebrates, often maintaining infections for life with no apparent ill-effects (59). Given that some of the viral species we identified are not only highly prevalent but also closely related to viruses previously discovered in tick cell lines (see Supplemental Note: *Rhabdoviridae*) (27), we next characterized several properties that could inform if and how these viruses are acquired and transmitted. Namely, we examined their distribution in ticks according to both life stages (larval, nymphal, and adult) and tissue localization (midgut and salivary glands). We performed multiplex PCR for the 13 viral sequences on cDNA generated from pools of salivary gland and midgut tissues dissected from an independent set of field-caught and laboratory-reared *I. pacificus* ticks of different life stages. Nine of the 13 viral sequences were detected in at least one of the dissected tissue pools across all life stages of the set of field or laboratory ticks (Fig. 3e).

Detection of virus in field larval ticks was particularly striking, since larval ticks have not yet consumed a bloodmeal, and ticks of any life stage are not known to transmit viruses directly within a population in the absence of a vertebrate host intermediate (60). Likewise, the detection of these viruses in laboratory-reared *I. pacificus* ticks indicated that the viruses can be maintained in the absence of natural bloodmeal hosts (Fig. 3; Fig. S4). These data suggest these viruses may be transmitted vertically within ticks, a phenomenon known to occur with several known tick viruses (60). Moreover, looking across life stages, we found that 8/9 viral sequences that we observed in the larvae were also detectable in either nymphs or adults, suggesting these viruses may be able

to persist or infect through multiple life stages. Our parallel analysis of the midgut and salivary tissues shows that the majority of the viral sequences (9/13) were also detectable in the salivary gland (Fig. 3e), the compartment of the tick generally associated with the potential for feeding-based (horizontal) transmission (Fig. 3d). While this does not definitively demonstrate that these viruses are transmissible by feeding, we hypothesize that this could be possible and worthy of further investigation.

## Identification of novel mRNA-like virus-like transcripts

In addition to the 13 viral RdRps, we identified 21 sequences with homology to an RdRp but with an open reading frame (ORF) structure inconsistent with known RNA viral genomes. Specifically, these RNA sequences encoded clusters of small ORFs with RdRp homology and large gaps (hundreds of bases) between their predicted ORFs (Fig. 4a). Many also contained multiple overlapping ORFs and/or small ORFs in opposite orientations. These unusual sequences were highly prevalent (Fig. 4a) and were independently assembled from multiple different ticks. Our analysis of these sequences suggested that they originated from many of the same families as the viral genomes (Fig. 4b), but with distinct sequences that encoded ORFs that were smaller and more numerous than would be expected for that family.

We were intrigued by the observed irregular genomic organization of these 21 virus-like sequences; thus we next sought to better understand their possible origins and functions. We first conducted experiments to eliminate potential artifactual explanations for the irregular ORF structure. To test whether these sequences could be the result of a misassembly, we selected one of the longest (5,043 bp) and most highly prevalent sequences (vlt_111) for more in-depth evaluation. We performed PCR with tiled overlapping primers (Fig. S6a) and applied 5′ and 3′ RACE to test if we could recover the vlt_111 sequence from the cDNA of a Garrapata tick (Materials and Methods). Sanger sequencing of the resulting PCR products confirmed the accuracy of our vlt_111 sequence assembly and indicated that vlt_111 is expressed as a 3′ poly-adenylated mRNA, ruling out the possibility that the non-canonical features of this RNA are due to misassembly of the original metatranscriptomic sequencing reads.

We also considered whether the irregular ORF structures we observed could be resolved with alternative codons, such as non-standard stop and start codons. We tested whether ORF prediction with any alternative genetic codes would result in an organization more consistent with that of a viral genome. Of the 25 known genetic codes tested, none substantially changed the ORF structure of any of the sequences (Fig. S6d), indicating that alternative genetic code alone cannot account for the observed genomic structure. Having eliminated possible artifactual explanations for how these sequences could originate from exogenous viral genomes, we termed these sequences of unknown origin and function "virus-like transcripts" (VLTs).

## VLTs likely serve a non-canonical function in *I. pacificus* ticks

Tick genomes are known to contain numerous endogenous viral elements (EVEs), which result from the horizontal integration of RNA viral sequences into tick genomes over the course of evolution (29). The observed high prevalence of VLTs led us to hypothesize that these mysterious sequences could have similar functions or histories. In the absence of a published genome assembly for *I. pacificus*, we could not check for corresponding sequences in a reference genome. Therefore, we developed PCR primers to screen for the VLTs in *I. pacificus* genomic DNA extracts isolated from the same field-collected ticks that were used for metatranscriptomic analysis (Fig. S6a). We confirmed the presence of vlt_111 in the field-collected and laboratory-reared tick DNA (Fig. S6b). To ensure this pattern was not specific to this particular VLT, we checked an additional five VLTs, all of which were present in lab-reared *I. pacificus* DNA (Fig. S6c), suggesting a genomic origin for these VLTs.

To confirm that the presence of DNA forms of transcripts is specific to the VLTs and not a general phenomenon, we additionally screened the genomic DNA for all of the

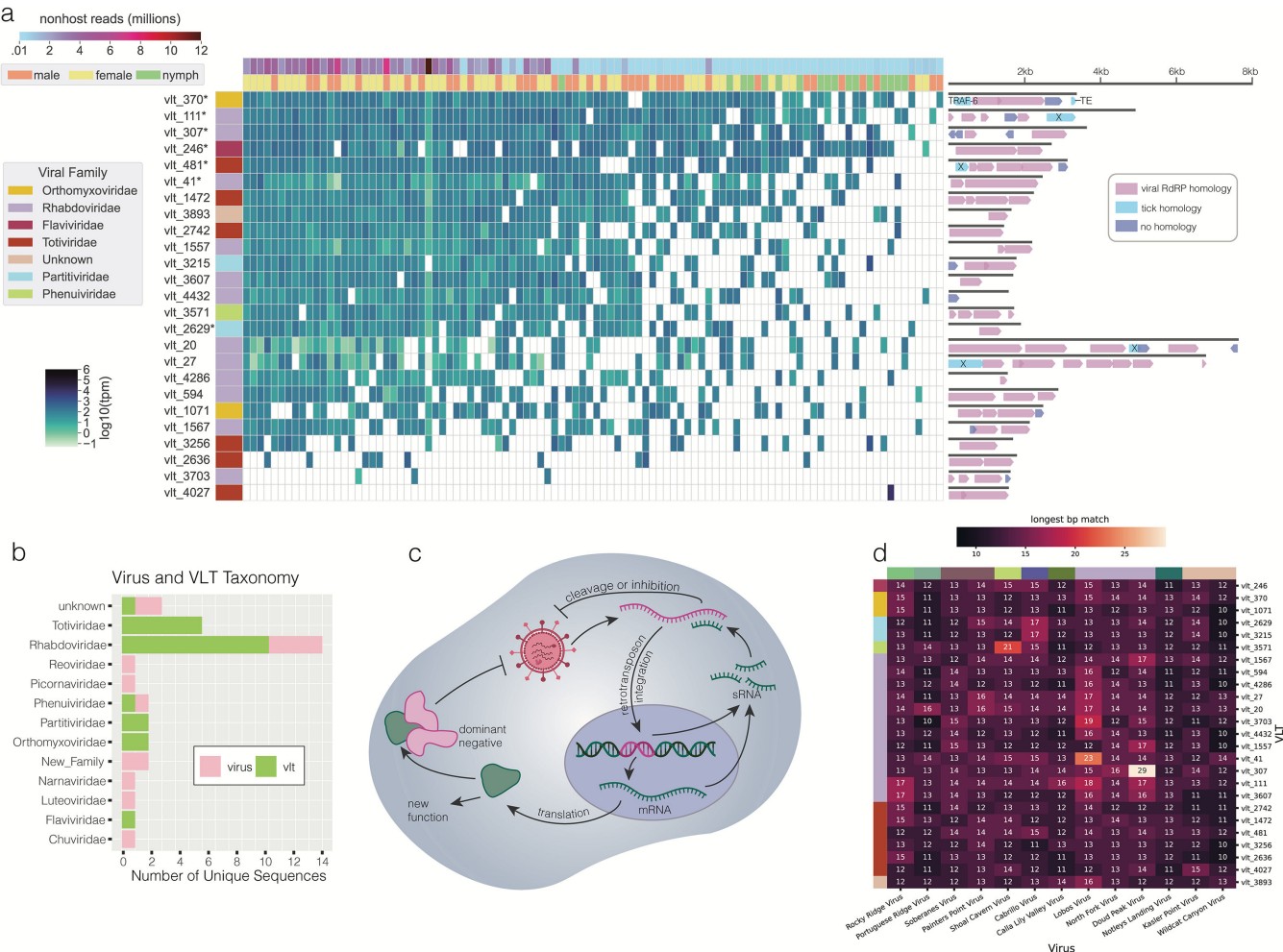

**FIG 4** Virus-like transcripts detected across *Ixodes pacificus* ticks in our study. (a) Heatmap showing transcripts per million (tpm) of virus-like transcripts (VLTs) across the data set. VLTs are colored by viral family of the closest blast hit and marked with "*" if they were detected in DNA by PCR. Predicted open reading frames (ORFs) are show to the right and colored by homology to viral RdRp (pink), tick proteins (turquoise; TRAF = TNF receptor-associated factor 6-like, TE = piggy Bac transposable element-derived protein 4-like, X = protein of unknown function), or no homology (blue). (b) Viral family assignment of exogenous viral sequences and VLTs. (c) Known functions of arthropod endogenous viral elements. (d) Heatmap displaying length of the longest perfectly matching sequences between each virus and each VLT. Rows and columns are colored by viral family.

presumed exogenous viral assemblies. Only one of the viral RNA genomes was detected in DNA. A faint band corresponding in size to Rocky Ridge virus was amplified from genomic DNA. This mivirus-like sequence could represent an intermediate between these two categories that is an exogenous RNA virus with a single or small number of recent genomic integrations into the *I. pacificus* genome. Alternatively, this could be caused by the production of DNA forms of the viral genome by endogenous retrotranscriptases (either prior to genomic integration or in the absence of integration) as in Salvati et al. (61). Given that its expression pattern mirrored that of the other exogenous viruses and its genome is both complete and contains the expected ORFs, we continued to classify this sequence as an exogenous virus and not a VLT.

To explore the possibility that our VLTs could have EVE-like functions, we next considered canonical pathways by which EVEs contribute to arthropod immunity (Fig. 4c). EVEs are most commonly known to function as non-coding RNAs. Much more rarely, they are expressed and translated as proteins that can act as dominant negative viral inhibitors or serve a new function (29, 30). As the fragmented ORF structure of our VLTs is inconsistent with the expression of full-length proteins, we focused on non-coding RNA functions. Typically, arthropod EVEs play antiviral roles by serving as a template for piwi

RNAs (piRNAs), 24–31 nucleotide (nt) RNAs that target an exogenous viral RNA genome for degradation by binding to a complementary sequence within it.

To test this model, we examined *in silico* whether our VLTs could give rise to small RNAs capable of binding the exogenous viral genomes in our data set through a matching sequence at least 24 nt long. Only one VLT contained a sequence of at least this length (vlt_307) matching one of our viral genome assemblies. Two others contained stretches longer than 20 nucleotides (Fig. 4d). The remaining 18 VLT sequences did not contain perfect matches longer than 17 nucleotides to any of the exogenous viruses. Of the three VLT-virus combinations with perfect matches of at least 20 nucleotides, none originated from the same tick sample (Fig. S6 f through h). In total, we did not uncover definitive evidence supporting the inhibition of viral replication through a canonical piRNA pathway. Our results point to either a non-canonical immunity mechanism or a different functional role entirely for the identified VLTs that could be explored in future studies.

Furthermore, we found evidence of this phenomenon beyond our own study. By examining the viral sequences deposited by Ni et al. (22) in a large metagenomic survey of the virome of ticks in China, we identified four sequences with an irregular ORF structure similar to that of our VLTs (Fig. S7a). Additionally, we used our VLTs to search for similar sequences in other tick genomes by BLAST. Most of the VLTs had either no homology to tick genomes or very short stretches of high homology (~100 bp). However, VLT 246, which has homology to segmented flaviviruses, also had significant homology to a Jingmen tick virus endogenous viral element in the *Ixodes ricinus* genome (Fig. S7b). These observations suggest that VLTs are features of tick genomes beyond our data set.

## DISCUSSION

In this study, we show the power of combining experimental enrichment of microbial sequences with single-tick metatranscriptomics for the identification of bacterial and viral transcripts in field *I. pacificus* tick communities. The ability to deeply sequence the non-host fraction allowed us to identify several genera of bacteria in *I. pacificus* previously unidentified in any tick species. Further investigation is warranted into the consequences and mechanisms underlying these symbiotic tick-bacteria partnerships.

Our approach also enabled us to uncover many novel viral RNA transcripts which we further investigated in a series of laboratory experiments. We found that several of these viral sequences were not only highly prevalent but were also present in tick salivary glands. This has important implications for public health, as bloodmeal hosts (including humans) are very likely to be exposed to viruses present in the salivary glands during feeding. Currently, tick-borne disease surveillance in California is focused on Lyme disease and a small number of other bacterial pathogens, but our results indicate humans in the region may also be exposed to diverse viruses (1). While our approach cannot determine whether any of these viruses are human pathogens, screening of clinical samples for exposure to them is a critical first step to understanding their disease-causing potential.

In addition to providing more comprehensive and quantitative insights into the *I. pacificus* microbiome, one of the most exciting and unexpected themes that emerged from our work relates to how viruses fit into the broader framework of the tick microbiome and tick biology. We found evidence that several viruses persisted in ticks across multiple life stages, including juvenile naïve larvae, as well as across field-caught and laboratory-reared populations, suggesting they are stable constituents of the tick microbiome. Certain viruses in *I. pacificus* may be able to establish and maintain independent niches within their tick hosts. These findings lay the groundwork for future work aimed at understanding tick–virus dynamics and how such relationships fit into tick physiology.

Further underscoring the critical importance of tick–virus interactions for *I. pacificus* biology was our discovery that numerous VLTs may originate from EVEs in the tick genome. Closer experimental evaluation of our VLTs pointed to a non-canonical

mechanism that is distinct from known antiviral pathways. While we cannot eliminate the possibility that VLTs are spuriously expressed without serving an adaptive function, EVEs are an underexplored feature of tick genomes, and future studies could determine whether VLTs represent a new mechanistic class of EVEs with adaptive contributions to tick immunity and biology.

The integration of RNA viral genomes into the tick genome as EVEs also provides a unique historical footprint for viruses that may have infected that tick host in the near or distant past. Interestingly, the VLTs identified in *I. pacificus* appear to derive from several viral families from which no exogenous viral genomes were found in this study, including the recently discovered segmented flaviviruses that cause febrile illness (62–64). Future field studies that expand on our *I. pacificus* virome analyses will help determine whether VLTs stem from ancient tick–virus interactions or contemporaneous interactions that were not captured in this study due to low abundance, limited sample size, and our focus on two collection locations. Altogether, our results highlight the need for more studies such as this in order to capture the full range of tic-associated microbes that could represent critical components of tick physiology or poorly understood pathogenic threats to human health. Our work provides an improved experimental and computational framework with increased sensitivity for low-abundance bacterial and viral taxa present in this increasingly important class of arthropod disease vectors.

## MATERIALS AND METHODS

### Tick sources

Ticks for metagenomic sequencing were collected by dragging from Garrapata State Park, California, USA ($n = 83$ adults) in December 2018 and from China Camp State Park, California, USA ($n = 17$ nymphs) in May 2019. Ticks for follow-up VLT PCR experiments were from the original China Camp collection. Ticks for tissue-tropism experiments were collected from Garrapata State Park in March 2022 ($n = 50$). Adult ticks were collected separately by sex. Ticks were stored in conical tubes according to groups (adult females, adult males, nymphs, and larvae) during transportation to the lab. Once in the lab, adult and nymphal ticks were surface sterilized in 1% bleach and frozen individually. Larval ticks were frozen as a pool and were not surface sterilized due to difficulties presented by their small size. All frozen ticks were stored at −80°C.

Laboratory-reared *I. pacificus* were received from the tick lab at the Centers for Disease Control and Prevention tick lab (Atlanta, GA, USA) and provided through BEI Resources (a service funded by the National Institute of Allergy and Infectious Diseases and managed by ATCC). Ticks were maintained in glass jars with a relative humidity of 95% (saturated solution of potassium nitrate) in a sealed incubator at 22°C with a light cycle of 16 h/8 h (light/dark).

### RNA extraction/library prep

Total RNA was extracted from the field-caught *I. pacificus* adult and nymph ticks in two separate batches. On ice, individual ticks were transferred to separate wells of a 96-well deepwell plate that was pre-loaded with a single 5 mm steel ball bearing (OMNI International, GA, USA) and 400 µL of 1× DNA/RNA shield (Zymo Research Corp., Irvine, CA, USA) in each well. The plates were sealed and subjected to bead bashing (3 × 3 min, with 1 min rest on ice in between each round of bashing) on a TissueLyser II beadmill (Qiagen, Valencia, CA, USA), then clarified by centrifugation at 2,000 rpm at 4°C for 5 min in a refrigerated tabletop centrifuge (Beckman Coulter, Indianapolis, IN, USA) to remove large debris. About 350 µL of the supernatant was transferred to a fresh 96-well deepwell plate and re-centrifuged under the same conditions to further clarify the homogenate. And 90 µL of the resulting supernatant was used as input for total RNA extractions; 110 µL of supernatant was transferred to a separate plate and archived at −80°C for potential follow-up analyses.

For both the adult tick and nymph tick homogenate preps, automated RNA extraction was performed in 96-well format (Bravo automated liquid handler, Agilent Technologies, Santa Clara, CA, USA) using a modified version of the Quick DNA/RNA pathogen magbead 96 extraction kit (Zymo Research Corp., Irvine, CA, USA) to automate total nucleic acid extraction and DNase treatment. RNA extracted from 90 µL of tick homogenates was eluted in a final volume of 25 µL of nuclease-free $H_2O$ into 96-well PCR plates. An aliquot of 3 µL was used for quantitative and qualitative analysis of the total RNA for each sample via Qubit fluorometer assay (Thermo Fisher Scientific, Waltham, MA, USA) and Agilent Bioanalyzer Pico 6000 total Eukaryotic RNA electrophoresis (Agilent Technologies, Santa Clara, CA, USA). A separate 5 µL aliquot was used as input for RNAseq library prep, and 2 × 7 µL aliquots were stamped into two separate daughter plates that were immediately frozen and archived at −80°C for potential follow-up studies.

The 96-well plates with 5 µL aliquots of adult tick and nymph tick total RNA preps were transferred to an automated liquid handler (Bravo automated liquid handler, Agilent Technologies, Santa Clara, CA, USA) for RNAseq library preparation in 96-well format. Briefly, the NEBNext Ultra II Non-directional RNAseq library preparation kit (New England Biolabs, Ipswich, MA, USA) was applied with the following modifications incorporated into the manufacturer's standard protocol: a 25 pg aliquot of External RNA Controls Consortium RNA spike-in mix ("ERCC," Thermo-Fisher, Waltham, MA, USA) was added to each sample prior to RNA fragmentation, and the input RNA mixture was fragmented for 8 min at 94°C prior to reverse transcription. A total of 12 cycles of PCR was performed using unique dual barcoded adapter primers for each sample at the amplification step. An initial equivolume pool was made up of 1 µL from each sample. Separate libraries were prepared for the adult and nymph ticks. SPRIselect (Beckman Coulter, Indianapolis, IN, USA) beads were used to size-select libraries with an average total length between 450 and 550 base pairs (bp). Library size distributions were verified by Agilent Bioanalyzer High Sensitivity DNA electrophoresis (Agilent Technologies, Santa Clara, CA, USA) and quantified by Qubit fluorometer (Thermo Fisher Scientific, Waltham, MA, USA). Paired-end 2 × 146 bp sequencing runs were separately performed on the adult and nymph equivolume pools with the Illumina MiSeq V2 Micro sequencing kit (Illumina, San Diego, CA, USA).

The yield of reads/µL acquired from the small-scale MiSeq run of the equivolume pools of individual libraries were used to normalize volumes of the individual adult tick and nymph tick libraries to generate approximately equimolar pools of samples for subsequent large-scale metatranscriptomic sequence analysis. The equimolar library pools were then subjected to depletion of highly abundant sequences (DASH) (41, 65), using a previously described pool of tick gRNAs (66) complexed with in-house prep of purified recombinant Cas9 protein. Resulting DASH'd libraries were qualitatively and quantitatively analyzed by Agilent Bioanalyzer High Sensitivity DNA electrophoresis (Agilent Technologies, Santa Clara, CA, USA) and Qubit fluorometer (Thermo Fisher Scientific, Waltham, MA, USA). While the DASH'd adult tick libraries provided sufficient material for large-scale metatranscriptomic sequencing, insufficient material remained in the tick nymph libraries that were DASH'd. Thus, for large-scale metatranscriptomic sequencing, the pool of DASH'd adult tick libraries was combined with a pool of un-DASH'd nymph tick libraries. This pooled prep was subjected to Paired-end 2 × 146 bp sequencing on the NextSeq2000 Illumina sequencing platform (Illumina, San Diego, CA, USA).

## Host subtraction/pre-processing and contig assembly

Fastq reads from the run were demultiplexed based on their associated dual barcode index sequences, then further processed using the CZID analysis pipeline (67). Libraries underwent quality filtering and adaptor trimming. Host reads were then removed by mapping to closely related genomes. Host-subtracted reads from each library were then assembled into contigs using SPADEs (68) within the CZID pipeline (67) version 3.7.

Both these resulting contigs and the host-subtracted reads were used for downstream analysis.

## Bacterial classification

Host-subtracted reads were classified using Kraken2 (version 2.1.1) (42). The full kraken2 database was used for classification. Only libraries with at least 1,000 classified reads were considered for analysis. Sequencing reads per taxon were converted to reads per million (rpm) using library size. To reduce false positives, the rpm value for each taxon was required to be at least 100 times the rpm in any of the control libraries (water and Hela cells) to be considered a positive. Additionally, at least 100 unique minimizers were required for each taxa. Taxa within each library not meeting these thresholds were excluded from the analysis. Fewer genera were detected in smaller libraries, including the nymphal ticks from China Camp SP (Fig. S1a), and samples clustered primarily by library size. We therefore focused subsequent analysis on libraries of at least one million non-host reads.

## Virus and virus-Like transcript identification

We focused our analysis on RNA viruses, as these tend to dominate arthropod viromes (33, 69, 70). Contigs were filtered to those of at least 1,500 bp in length. ORFs were predicted for these contigs using prodigal (71), and the resulting proteins were searched using HMMscan from HMMER3 (version 3.3.2) (58) against a collection of HMMR profiles of viral RdRps. The following RdRp HMMs were downloaded from the pfam database (72) on 4 March 2021; RdRP_1, RdRP_2, RdRP_3, RdRP_4, RdRP_5, Viral, RdRp_C, Mitovir_RNA_pol, Mononeg_RNA_pol, Birna_RdRp, and Bunya_RdRp. Additionally, custom HMMs were constructed from RdRp sequences for members of *Narnaviridae* and *Orthomyxoviridae* (sequences and combined HMM available in supplement).

All putative RdRp hits from HMMER were queried against the full NCBI nonredundant protein database (as of 24 January 2021) using diamond blastp version 0.9.24 to identify their closest hit (73). For putative RdRps with multiple hits, the alignment with the highest bitscore was reported. Putative RdRp sequences covering less than 30% of their closest blast hit were initially classified as putative virus-like transcripts, while those above that threshold were classified as exogenous viral sequences. The ORF structures of all contigs with RdRp sequences were manually inspected to confirm this classification. Those that showed an ORF structure similar to canonical viral genome sequences were classified as exogenous viral genome sequences; while those that contained significant gaps between ORFs or multiple reading frames with RdRp homology (where one was expected) were classified as VLTs.

To identify additional segments of multipartite viral genomes, we searched for contigs that strongly co-occurred with the contigs encoding an RdRp. Presence/absence of potential RdRp contig clusters in each library was based on evidence of quality- and host-filtered reads mapped to each contig. Presence was coded as 1 and absence as 0, and the Jaccard distance was calculated for all pairs of contigs. Any sequence with Jaccard distance <0.4 was classified as a putative additional genomic segment (Fig. S5). In our determination of whether a putative additional genomic segment represented a potential segment for its co-occurring RdRp contig cluster, we further considered (i) whether additional segments are expected for the viral family of the given RdRp contig cluster, and (ii) the sequence similarity (if any) of each putative additional genome segment to viral sequences in the NCBI databases. If additional segments were expected the contig was considered a segment originating from the same genome as the RdRp contig.

To infer the level of virus phylogeny to which each of the recovered exogenous viral sequences could be mapped, we downloaded the top 100 blast matches for each of the RdRp protein sequences that we recovered to create a representative set of related viral RdRp sequences for comparative sequence analysis. The downloaded sequences spanned the following viral taxa: *Bunyavirales*, *Mononegavirales*, *Reovirales*,

*Picornaviridae*, *Narnaviridae*, *Solemoviridae*, and *Chuviridae*. Within each of these taxa, we used CD-HIT (74) version 4.8.1 to filter the resulting initial sets of 100 RdRp sequences matches using a threshold of 85% nucleotide identity, then used MAFFT (75) version 7.475 to align each of the putative viral RdRp protein sequences with their corresponding taxonomic order of representative set of RdRp protein sequences. Maximum likelihood trees were estimated using iqtree 2.0.3 (76), using the model finder limited to all viral substitution models and with 1000 bootstraps. Resulting trees were visualized in iTOL version 5 (77) (Fig. S8).

## Determination of prevalence and abundance

Assembled contigs were clustered using cd-hit-est (CD-HIT version 4.8.1) at a threshold of 85% nucleotide identity (74). Chuvirus genomes were rotated to a common start position using a custom python script to ensure accurate clustering. 85% nucleotide identity was chosen as a cutoff to minimize multi-mapping reads between closely related sequences. However, some clusters contained significant sequence diversity and could potentially be considered to contain multiple species. The representative sequences from this clustering were used for all downstream analyses.

Reads from each sequencing library were mapped back to the collection of cluster representatives using bowtie2 version 2.4.1 (78). Sequences aligning to each contig were counted using samtools idxstats version 1.9 (79). Aligned reads, contig length, and library size were used to calculate rpm and transcripts per million (tpm) values for each library. To consider a contig "present" in a given library, the rpm value was required to be greater than 10 times the value in any of the control libraries. This filter was designed to remove potential false positives caused by cross-contamination of high-titer species.

## Rarefaction analysis

The viral genomes in each sample were determined by the presence of a contig of at least 1,000 base pairs (bp) that clustered with one of the 13 representative genomes identified. The presence of a contig rather than read mapping was used to simulate viral discovery in each sample, under the assumption that new viruses discovered may be too divergent to detect by read mapping. The samples were ordered by the number of new genomes seen (not seen in any of the previous samples). The number of new genomes was counted for the addition of each sample. This was repeated for 50 iterations and the median number of new samples at each step was determined. The Chao index was calculated using the R library fossil version 0.40 (80, 81) .

## Co-occurrence of taxa

To determine whether any pairings of taxa (either bacterial or viral) occur more or less frequently than expected given their prevalence, we utilized the recently developed metric $\alpha$(48). The presence of each taxon was considered at the genus level for bacteria and at the species level for viruses. The distance $\alpha$ and associated $P$ value were determined for all pairs of taxa using the CooccurrenceAffinity R package (version 1.0) (48). Pairs were filtered to those with a $P$ value $\leq 0.005$. These relationships were visualized as a network with edges corresponding to $\alpha$ and nodes corresponding to taxa using the R package igraph (version 1.3.0) (82). Node size was scaled according to taxon prevalence in the data set.

## Virus-like transcript mining

We used the bioinformatics methodology from this study to search for the presence of VLTs in the contigs from Ni et al. (22), a metagenomic data set of ticks from China. The open reading frame structure of all of the deposited contigs were visually examined using geneious for the same types of patterns as observed in our data set, including early stop codons, overlapping reading frames, multiple reading frames with RdRp homology.

To determine whether the VLTs observed in our data set are present in other tick genomes, we compared the 25 representative VLT nucleotide sequences to tick genomes using BLAST. BLAST was performed using the NCBI web server on 19 December 2023, employing the blastn algorithm with word size of 11, *e*-value of 0.05, and limiting results to taxid 6939 (*Ixodidae*).

## Virus-like transcript PCR

To verify that VLTs were present in tick genomes and expressed in field-caught and laboratory-reared ticks, we extracted tick RNA and genomic DNA and confirmed the presence of VLTs with PCR. Adult male *I. pacificus* ticks from China Camp (*n* = 3) were pooled and homogenized by beating for two increments of 30 s at 4,000 bpm in a bench homogenizer (Bead Bug, Benchmark Scientific) with 1.4 mm zirconium oxide ceramic beads (Fisher Scientific) in ice-cold TRIzol reagent (Thermo Fisher Scientific). RNA extraction was performed using a Zymo Research Direct-zol RNA Microprep kit (Zymo Research), and RNA was converted to cDNA using Primescript RT reagent kit (Takara Bio) in 10 µL reactions using random hexamer primers and following manufacturer protocols. Genomic DNA was extracted from adult male *I. pacificus* ticks (*n* = 3, separate individuals from RNA), which were pooled, flash-frozen, and ground to powder. A DNeasy Blood & Tissue Kit (Qiagen) was used to extract genomic DNA following the manufacturer's protocols.

3′ RACE was conducted using the 3′ RACE system for rapid amplification of cDNA ends kit (Thermo Fisher Scientific) according to the manufacturer's protocols. Nested PCR using VLT-specific primers was used to amplify regions of VLTs from tick cDNA and genomic DNA in two steps using 10 µM VLT-specific primers and the UAP primer provided in the kit. Nested PCR primer sequences were designed by hand, following the protocol from the kit for the 3′ RACE system for rapid amplification of cDNA ends. We verified the absence of hairpin loops with IDT oligoanalyzer, and then we used NCBI BLAST to verify that there were no non-specific targets. The resulting primer sequences can be found in Table S2. We loaded 5 µL of product onto 0.7% agarose gels and ran them at 160 V for 2 h before imaging. PCRs were purified using a QIAquick PCR purification kit (Qiagen) and unidirectionally Sanger sequenced (Genewiz) with the nested PCR primer.

## Virus multiplex PCR

Viral genome sequences were analyzed with Snapgene (Dotmatics) and PrimerPlex software (Premier Biosoft) to design four sets of multiplex primers that amplify 100–550 bp regions. Platinum SuperFi II Green PCR Master Mix (Thermo Scientific) was used for all PCR reactions. Mixed cDNA from the original sequenced field-collected ticks was used as a positive control. No-template (water only) reactions were also included as negative controls. Primer pair sequences are listed in Table S2. PCR reactions were analyzed by electrophoresis using a 2% agarose gel containing GelRed (Biotium) and visualized using an Azure c400 imager (Azure Biosystems). At least one band corresponding to each virus was cut out of the agarose gel, purified using the QIAquick Gel extraction kit (Qiagen), and Sanger sequenced (GeneWiz) to confirm correct PCR specificity.

## Tick dissections/extractions

For tissue-tropism determination, wild-collected ticks (*n* = 20) from Garrapata State Park were dissected using a micro scalpel cleaned with 70% isopropanol and a sterile needle. Tissues were pooled from multiple ticks rather than analyzed individually as it was not possible to obtain sufficient RNA from the midgut or salivary glands of a single tick. Males and females were processed separately and each pool of tissues contained material from 3 to 10 individuals.

Ticks were pooled. The scalpel was cleaned with 70% isopropanol and the needle was replaced between pools. The tick cuticle was excised and the midgut and salivary

glands were removed using tweezers cleaned with 70% isopropanol (cleaned between each pool). Tissues were pooled and rinsed in droplets of phosphate-buffered saline, then transferred by pipette into 300 µL of TRIzol. PCR results from males and females, as well as biological replicates are collapsed for simplicity in Fig. 3d, but sex-specific and replicate-specific results can be found in Fig. S4d.

Whole adult ticks were added to 300 µL of TRIzol in pools of four to five individuals, grouped by sex. Nymphal ticks were added to TRIzol in a pool of 3, and larval ticks were flash-frozen and added to TRIzol in pools of 10–15. All ticks and tick tissues were homogenized by bead-beating with ceramic beads in increments of 30 s. Samples were placed on ice between cycles and cycles continued until tissue was visually homogenized.

RNA extraction was performed using Directzol RNA Extraction kits, with on-column DNAse1 treatment. RNA was converted to single-stranded cDNA using Quantabio cDNA mastermix in 10 µL reactions.

## ACKNOWLEDGMENTS

We would like to thank Anne Sapiro, Gytis Dudas, Kishen Patel, and Raul Andino for their helpful conversations and feedback on this project. We are grateful to Greg Huber, Olga Botvinnik, and Norma Neff at Chan Zuckerberg Biohub San Francisco for their strategic and technical input. We also thank members of both the Chou lab and the lab of Andrea Swei for assistance collecting ticks in the field. We greatly appreciate all members of the Chou and Pollard labs for their input and feedback on experiments, analyses, and manuscript preparation.

## AUTHOR AFFILIATIONS

[1]Department of Biochemistry & Biophysics, University of California–San Francisco, San Francisco, California, USA

[2]Gladstone Institute of Data Science & Biotechnology, San Francisco, California, USA

[3]One Health Institute, Colorado State University–Fort Collins, Fort Collins, Colorado, USA

[4]Department of Biomedical Sciences, Colorado State University–Fort Collins, Fort Collins, Colorado, USA

[5]Chan Zuckerberg Biohub, San Francisco, San Francisco, California, USA

[6]Department of Biology, Colorado State University–Fort Collins, Fort Collins, Colorado, USA

[7]Department of Epidemiology & Biostatistics, University of California San Francisco, San Francisco, California, USA

## AUTHOR ORCIDs

Calla Martyn http://orcid.org/0000-0002-1886-5198
Amy Kistler http://orcid.org/0000-0003-1112-719X
Katherine S. Pollard http://orcid.org/0000-0002-9870-6196
Seemay Chou http://orcid.org/0000-0002-7271-303X

## FUNDING

| Funder | Grant(s) | Author(s) |
| --- | --- | --- |
| Chan Zuckerberg Biohub | | Calla Martyn |
| | | Domokos Lauko |
| | | Gloria Castaneda |
| | | Amy Kistler |
| | | Katherine S. Pollard |
| Gladstone Institutes (J. David Gladstone Institutes) | | Calla Martyn |

| Funder | Grant(s) | Author(s) |
|--------|----------|-----------|
| | | Katherine S. Pollard |
| HHS | National Institutes of Health (NIH) | R01 AI32851 | Beth M. Hayes |
| Pew Charitable Trusts (PCT) | | Beth M. Hayes |

## AUTHOR CONTRIBUTIONS

Calla Martyn, Conceptualization, Data curation, Formal analysis, Investigation, Methodology, Validation, Visualization, Writing – original draft, Writing – review and editing | Beth M. Hayes, Validation, Writing – original draft, Writing – review and editing | Domokos Lauko, Validation, Writing – original draft | Edward Midthun, Investigation, Validation | Gloria Castaneda, Investigation, Methodology | Angela Bosco-Lauth, Funding acquisition, Investigation, Supervision | Daniel J. Salkeld, Investigation, Methodology | Amy Kistler, Conceptualization, Data curation, Investigation, Methodology, Resources, Supervision, Validation, Writing – original draft, Writing – review and editing | Katherine S. Pollard, Funding acquisition, Investigation, Methodology, Project administration, Resources, Supervision, Writing – original draft, Writing – review and editing | Seemay Chou, Conceptualization, Funding acquisition, Investigation, Project administration, Resources, Supervision, Writing – original draft, Writing – review and editing

## DATA AVAILABILITY

Sequencing data are available at the NCBI Sequence Read Archives under project PRJNA870442. Intermediate files, including assembled contigs and read coverage can be found on figshare at https://figshare.com/projects/I_pacifics_mNGS/144081. Viral assemblies have been deposited at NCBI GenBank under accession numbers PP415825, PP415826, PP415827, PP415828, PP415829, PP415830, PP415831, PP415832, PP415833, PP415834, PP415835, PP415836, PP415837, PP415838, PP415839, PP415840, PP415841, PP415842, PP415843, PP415844, PP415845, PP415846, PP415847, PP415848, and PP415849. In addition, they can be accessed at FigShare using DOI 10.6084/m9.figshare.20497227. *I. pacificus* genome sequence (released February 2018) is available at s3.us-west-2.amazonaws.com: https://czid-public-references.s3.us-west-2.amazonaws.com/host_filter/ticks/2018-02-15-utc-1518652800-unixtime__2018-02-15-utc-1518652800-unixtime/tick_genomes.fa and as a supplemental data file. Analysis notebooks and scripts can be found on github at https://github.com/callamartyn/ipac_virus. CZID pipeline results, including intermediate host subtracted reads, assemblies, and other intermediate files can be found at https://czid.org/pub/uuFkacq3hT (Garrapata adults), https://czid.org/pub/EEJfJPNhYP (Garrapata adults, resequencing of low-coverage samples), and https://czid.org/pub/ss3AnxpDbU (China Camp nymphs).

## ADDITIONAL FILES

The following material is available online.

### Supplemental Material

**Supplemental Figures (mSystems00321-24-S0001.pdf).** Figures S1-S8.
**Supplemental Note (mSystems00321-24-S0002.docx).** Descriptions of viral genome assemblies.
**Table S1 (mSystems00321-24-S0003.docx).** Summary of the viruses identified.
**Table S2 (mSystems00321-24-S0004.docx).** All PCR primer sequences used.

### Open Peer Review

**PEER REVIEW HISTORY (review-history.pdf).** An accounting of the reviewer comments and feedback.

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
