## [Reviewer comments · mSystems]

Metatranscriptomic Investigation of Single *Ixodes pacificus* Ticks Reveals Diverse Microbes, Viruses, and Novel mRNA-like Endogenous Viral Elements

Calla Martyn, Beth Hayes, Domokos Lauko, Edward Midthun, Gloria Castaneda, Angela Bosco-Lauth, Daniel Salkeld, Amy Kistler, Katherine Pollard, and Seemay Chou

Corresponding Author(s): Katherine Pollard, Gladstone Institutes

Review Timeline:

Submission Date:

March 9, 2024

Accepted:

March 27, 2024

Editor: Jonathan Klassen

Reviewer(s): The reviewers have opted to remain anonymous.

Transaction Report:

DOI: <https://doi.org/10.1128/msystems.00321-24>

Re: mSystems00321-24 (Metatranscriptomic Investigation of Single *Ixodes pacificus* Ticks Reveals Diverse Microbes, Viruses, and Novel mRNA-like Endogenous Viral Elements)

Dear Dr. Katherine S. Pollard:

Your manuscript has been accepted, and I am forwarding it to the ASM production staff for publication. Your paper will first be checked to make sure all elements meet the technical requirements. ASM staff will contact you if anything needs to be revised before copyediting and production can begin. Otherwise, you will be notified when your proofs are ready to be viewed.

There is one small issue that I noticed during my final read of the manuscript that I suggest could be addressed during the proof stage: L. 136 you may wish to replace "Candidatus" with "Candidatus Carsonella".

Cover Image Submissions: If you would like to submit a potential Cover Image, please email a file and a short legend to msystems@asmusa.org. Please note that we can only consider images that (i) the authors created or own and (ii) have not been previously published. By submitting, you agree that the image can be used under the same terms as the published article. Image File requirements: TIF/EPS, 7.5 inches wide by 8.25 inches tall (at least 2,250 pixels wide by 2,475 pixels tall), minimum 300 dpi resolution (600 dpi preferred), RGB, and no figure elements, e.g., arrows or panel labels. The legend should be a short description of the image, 1-2 sentences recommended.

We recognize that the video files can become quite large, so to avoid quality loss ASM suggests sending the video file via <https://www.wetransfer.com/>. When you have a final version of the video and the still ready to share, please send it to mSystems staff at msystems@asmusa.org.

Sincerely,

Jonathan Klassen
Editor
mSystems

Reviewer #1 (Comments for the Author):

I enjoyed reading the revised version of this manuscript and I recommend publication of this manuscript as it is.